# Factors associated with aspirin resistance in diabetic patients: A metabolic and inflammatory profile analysis

Bo Chen[1☉], Zisheng Li[1☉], Jianyong Zhao[2☉], Huamei Dong[2☉], Long Tong[2], Jiaqing Dou[2*]

1 Department of Nuclear Medicine, Chaohu Hospital Affiliated to Anhui Medical University, Hefei, Anhui, China, 2 Department of Endocrinology, Chaohu Hospital Affiliated to Anhui Medical University, Hefei, Anhui, China

☉ These authors contributed equally to this work.
* djqch@163.com

## Abstract

### Background

Diabetes mellitus (DM) is strongly linked to both first-time and recurrent atherosclerotic thrombotic events. Although aspirin (ASA) is commonly used to prevent cardiovascular diseases, studies have shown that ASA does not significantly reduce the risk of cardiovascular events in DM patients. This inconsistency highlights the need for further research into the underlying mechanisms of ASA resistance. Therefore, this study investigates the factors associated with aspirin resistance in DM patients, aiming to offer insights for improving cardiovascular disease prevention in this group. This study specifically investigated biochemical aspirin resistance, defined as inadequate suppression of thromboxane biosynthesis.

### Methods

This prospective case-control study enrolled 53 DM patients and 66 age-/sex-matched healthy controls. Baseline metabolic-inflammatory markers—including BMI, LDL-C, cystatin C (CysC), hs-CRP, and HOMA-IR—were assessed alongside urinary 11-dehydrothromboxane B2 (11dhTxB2) levels pre- and post-aspirin intervention (81–100 mg/day × 7 days). Biochemical aspirin resistance was defined as post-administration urinary 11dhTxB2 ≥ 1500 pg/mg creatinine, reflecting inadequate suppression of total body thromboxane biosynthesis. Group comparisons utilized nonparametric tests (Mann-Whitney U) for skewed variables and $\chi^2$ tests for categorical data. The influencing factors of ASA resistance were investigated through univariate analysis and logistic regression analysis, with multiple linear regression analysis being applied to model the Δ11dhTxB2 (post- vs. pre-administration difference).

**Data availability statement:** All relevant data are within the manuscript and its Supporting information files.

**Funding:** This study was supported by the Anhui Medical University School Fund Grant (2023xkj069). No additional external funding was received for this study. The funders had no role in the study design, data collection and analysis, decision to publish, or preparation of the manuscript.

**Competing interests:** The authors have declared that no competing interests exist.

## Results

Compared to age-/sex-matched controls, diabetic patients exhibited significantly elevated metabolic-inflammatory markers (BMI, LDL-C, CysC, hs-CRP, HOMA-IR; all $P < 0.01$) and 60% higher baseline urinary 11dhTxB2 levels (2,324.58 vs. 1,452.51 pg/mg creatinine; $P = 0.001$), with persistent post-ASA disparity (1,205.31 vs. 1,058.02 pg/mg creatinine; $P = 0.007$). Biochemical aspirin resistance prevalence was 2.7-fold higher in diabetes (20.8% [11/53] vs. 7.6% [5/66]; $P = 0.036$). Univariate analysis linked Pre-11dhTxB2,higher BMI, hs-CRP, and HOMA-IR to ASA resistance (all $P < 0.05$), though multivariable logistic regression showed nonsignificant trends. Logistic regression analysis revealed that each unit increase in baseline 11dhTxB2 was associated with a 0.2% increase in the odds of aspirin resistance. Multivariable linear regression identified systemic inflammation (hs-CRP: $B = 2,147.6$, $P < 0.001$) and higher BMI (BMI: $B = 204.9$, $P = 0.021$) were strongly associated with attenuated thromboxane suppression (Δ11dhTxB2).

## Conclusion

Patients with diabetes exhibit heightened thromboxane biosynthesis and a markedly elevated prevalence of biochemical aspirin resistance compared to healthy individuals, underscoring a prothrombotic phenotype linked to metabolic-inflammatory dysregulation. Higher BMI and systemic inflammation emerged as key factors associated with attenuated aspirin efficacy, suggesting platelet activation pathways beyond conventional COX-1 inhibition or involving non-platelet sources. Early identification of platelet hyperreactivity, coupled with targeted metabolic control and anti-inflammatory strategies, may refine personalized cardiovascular prevention in this high-risk population,while acknowledging that persistent urinary 11dhTxB2 elevation post-aspirin likely reflects significant non-platelet thromboxane generation.

## Introduction

Diabetes mellitus (DM) is a major risk factor for both initial cardiovascular events and the subsequent worsening of conditions [1]. Individuals with diabetes have a significantly higher risk of cardiovascular (CV) diseases compared to non-diabetic patients with similar characteristics, primarily due to metabolic and functional changes in vascular cells [2]. This disproportionate risk is not only attributed to the direct impact of diabetes on blood glucose regulation, but also to its exacerbation of the cardiovascular system's burden through pathophysiological mechanisms such as oxidative stress, chronic low-grade inflammation, dyslipidemia, and insulin resistance. These factors increase susceptibility to severe cardiovascular events like myocardial infarction and stroke [3].

Antiplatelet agents, such as aspirin (ASA), ticagrelor, and clopidogrel, are widely used to inhibit platelet function and lower the risk of atherosclerotic thrombosis [4].

Given the elevated cardiovascular risk in diabetic patients, antithrombotic therapy could provide greater clinical benefits for primary and secondary prevention [5]. However, the benefit-risk balance of aspirin in primary cardiovascular prevention remains controversial. Three large randomized, placebo-controlled trials in diverse populations, including healthy elderly individuals and diabetic patients, have been conducted. Results indicated that aspirin did not significantly lower cardiovascular event risk in diabetic patients, and its mechanism requires further investigation [6,7]. Due to insufficient evidence supporting aspirin's widespread use, international guidelines do not advocate its routine use in diabetic patients without atherosclerotic cardiovascular disease (ASCVD). Instead, individualized treatment strategies are recommended to balance aspirin's protective effects against ischemic events with bleeding risk [8].

Aspirin irreversibly acetylates cyclooxygenase-1 (COX-1), blocking the conversion of arachidonic acid into thromboxane A2 (TxA2) and prostacyclin (PGI2), thus inhibiting platelet activation and aggregation [9]. TxA2, a biologically active thromboxane metabolite, is clinically significant, making its detection crucial. However, TxA2's high instability and short half-life (20–30 seconds) make conventional measurement technically difficult. Therefore, urinary 11-dehydro-thromboxane B2 (11dhTxB2), a stable enzymatic metabolite of TxA2, serves as a biomarker for total body TxA2 biosynthesis [10]. In the absence of aspirin therapy, it reflects contributions from both platelets and other cellular sources (e.g., monocytes, endothelium). Following aspirin administration, which irreversibly inhibits platelet COX-1, persistently elevated urinary 11dhTxB2 levels are thought to predominantly reflect TxA2 biosynthesis from non-platelet sources (e.g., via COX-2 or rapidly regenerated COX-1 in nucleated cells) [11,12], although the relative contributions pre-therapy in conditions like diabetes remain incompletely defined. Despite this complexity, elevated urinary 11dhTxB2 levels, particularly post-aspirin, have been consistently associated with increased cardiovascular risk [13].

Evidence suggests that premature and widespread vascular lesions in diabetes patients, coupled with a prothrombotic environment, contribute to platelet hyperactivity, which plays a key role in cardiovascular outcomes [14]. Oxidative stress, dyslipidemia, and diabetes-specific factors like insulin resistance and hyperglycemia are key contributors to platelet dysfunction [15]. Based on this, we hypothesize that diabetes-related metabolic states, such as hyperglycemia, insulin resistance, low-grade inflammation, obesity, dyslipidemia, and renal insufficiency, promote platelet activation, resulting in increased urinary 11dhTxB2 levels. Therefore, this study aims to assess the association between urinary 11dhTxB2 levels and metabolic parameters in diabetes patients, further exploring potential mechanisms underlying poor aspirin response in this population. Specifically, we investigate biochemical aspirin resistance, defined as persistent elevation of urinary 11dhTxB2 post-aspirin administration, acknowledging its reflection of total body thromboxane biosynthesis with likely significant non-platelet contributions under therapy.

## Methods

### Study design and participants

This study employed a prospective cohort design with matching. A total of 53 patients with type 2 DM were recruited from the outpatient and inpatient departments of Chaohu Hospital Affiliated to Anhui Medical University. All participants were diagnosed according to internationally recognized diagnostic criteria for diabetes. The control group consisted of 66 age- and sex-matched healthy individuals, who were selected from hospital staff and local community members, all of whom had normal blood glucose levels. The study period extended from February 1 to December 31, 2024. All participants provided written informed consent, and the study protocol was approved by the Ethics Committee of Chaohu Hospital Affiliated to Anhui Medical University (approval number: KYXM202401001). The study was conducted in accordance with the Declaration of Helsinki.

**Inclusion criteria.** Participants were screened by the attending physician based on internationally recognized diagnostic criteria for diabetes, specifically defined by at least one of the following metabolic markers: fasting plasma glucose (FPG) levels ≥ 7.0 mmol/L, 2-hour postprandial blood glucose ≥ 11.1 mmol/L during an oral glucose tolerance test (OGTT), or elevated glycated hemoglobin (HbA1c) ≥ 6.5%, confirmed through laboratory testing.

**Exclusion criteria.** Pregnant or lactating women; individuals allergic to or intolerant of enteric-coated aspirin; those with organic gastrointestinal diseases, such as gastrointestinal tumors, peptic ulcers, or inflammatory bowel disease; individuals with severe dysfunction of the heart, brain, liver, or kidneys (e.g., NYHA Class III/IV heart failure), or other major systemic diseases; current or recent use (within the preceding 3 months) of antithrombotic medications other than aspirin; and individuals with a documented history of major cardiovascular diseases (e.g., myocardial infarction, stroke, or coronary revascularization).

**Data collection.** Blood samples were collected from all participants after an overnight fast (≥8 hours) to minimize diurnal and dietary variability. Venous blood was drawn into EDTA-anticoagulated tubes for plasma separation and serum-separating tubes (SST) for serum biomarkers. Plasma and serum were isolated by centrifugation at 3,000 rpm for 10 minutes within 30 minutes of collection. Aliquots were stored at −80°C until analysis to prevent analyte degradation. Hemolyzed or lipemic samples were excluded based on visual inspection. Data included the following: demographic data (age, gender); anthropometric parameters (body mass index, BMI);glucose metabolism biomarkers: glycated serum protein (GSP), homeostatic model assessment for insulin resistance (HOMA-IR) index, calculated as [fasting insulin (µU/mL) × fasting glucose (mmol/L)]/22.5; lipid markers (low-density lipoprotein cholesterol, LDL-C); renal function biomarkers (cystatin C, Cysc); systemic inflammation markers (high-sensitivity C-reactive protein, hs-CRP); and baseline urinary 11dhTxB2 levels.

**Intervention.** Following baseline assessments, both the experimental group (DM patients) and the control group (healthy individuals) received enteric-coated low-dose aspirin (81–100 mg/day) orally once daily for 7 consecutive days. Urine samples for 11dhTxB2 measurement were collected 24 hours after the last aspirin intake to standardize the timing of sample collection relative to the final dose.

## 11dhTxB2 quantification and ASA response assessment

Urinary 11dhTxB2 levels were measured using chemiluminescence immunoassay (CLIA; RangeCL-300i analyzer), while urinary creatinine concentrations were determined via enzyme-linked immunosorbent assay (ELISA). To simplify sample collection protocols, randomly collected urine samples were utilized based on established evidence that urinary 11dhTxB2 levels exhibit a strong correlation with creatinine-adjusted values (r > 0.85 in prior validation studies) [16], eliminating the need for 24-hour urine collection. The 11dhTxB2 concentration was calculated using a standard curve generated from a reference solution, with results normalized to urinary creatinine and expressed as pg 11dhTxB2 per mg creatinine (pg/mg).

Participants were classified as good biochemical responders (post-aspirin urinary 11dhTxB2 < 1500 pg/mg creatinine) or poor biochemical responders (exhibiting biochemical aspirin resistance/non-response) (≥ 1500 pg/mg creatinine) [13,17,18]. This threshold of 1500 pg/mg creatinine was selected based on the manufacturer's validation data for the specific assay used in this study [13,18], which established it through analytical comparison to a first-generation assay with an arbitrary threshold. We acknowledge this cutoff lacks robust clinical outcome validation. Recent studies report variable outcome associations with different thresholds for this assay [19]. Therefore, our definition reflects assay-specific biochemical non-suppression.

Biochemical aspirin resistance, defined here as persistent elevation of urinary 11dhTxB2 (≥ 1500 pg/mg creatinine) despite therapy, primarily signifies persistent systemic thromboxane biosynthesis. It is crucial to note that while pre-aspirin urinary 11dhTxB2 reflects total body TxA2 production from both platelet and non-platelet sources, post-aspirin levels in compliant individuals predominantly reflect TxA2 generation from non-platelet sources (e.g., vascular endothelium, monocytes via COX-2 or rapidly resynthesized COX-1) [17]. Therefore, this biochemical resistance likely originates extra-plateletly, rather than solely indicating a failure to inhibit platelet COX-1. While this specific cutoff has been employed in previous research to distinguish responders from non-responders, we acknowledge it is assay-specific and not a universal consensus criterion. Although direct associations between this exact threshold and clinical outcomes are limited, elevated

urinary 11dhTxB2 levels have been consistently linked to increased cardiovascular risk in multiple studies [13,17,18]. The study specifically investigates this biochemical aspirin resistance/non-response and its association with metabolic dysregulation in DM.

### Statistical methods

Statistical analyses were performed using IBM SPSS Statistics 25.0. Continuous variables with skewed distributions (assessed via Shapiro-Wilk test, $P < 0.05$) were expressed as median and interquartile range (IQR) and compared between DM and control groups using the Mann-Whitney U test. Categorical variables were presented as numbers (%) and analyzed with the $\chi^2$ test or Fisher's exact test (if expected cell counts <5). Potential factors associated with aspirin resistance in patients with DM were preliminarily screened via univariate analysis. Independent predictors were subsequently determined through multivariable logistic regression, with adjusted ORs and 95% CIs reported. Additionally, multivariable linear regression adjusted for age, sex, and baseline metabolic markers was used to evaluate independent determinants of Δ11dhTxB2 (post- vs. pre-administration difference). To evaluate within-subject changes in Δ11dhTxB2 levels, we utilized the Wilcoxon signed-rank test. All tests were two-tailed, with statistical significance defined as $P < 0.05$.

## Results

### Comparative analysis of metabolic profiles, inflammatory markers, and ASA response phenotypes

This study compared metabolic, inflammatory, and aspirin resistance-related indicators between DM patients and healthy controls. The results indicated no significant differences in age and gender distribution between the two groups ($P > 0.05$). However, DM patients exhibited significantly elevated levels of BMI, LDL-C, CysC, hs-CRP, and HOMA-IR compared to the control group (all $P < 0.05$).

Aspirin intervention reduced urinary 11dhTxB2 levels by an average of 41.39% in DM patients, compared to a 31.12% reduction in healthy controls; however, the intergroup difference in suppression magnitude did not reach statistical significance ($P = 0.067$). As illustrated in Fig 1, the distribution of 11dhTxB2 level changes pre- and post-ASA administration demonstrates higher baseline thromboxane biosynthesis in DM patients (pre-ASA: 2324.58 vs. 1452.51 pg/mg creatinine; $P = 0.001$), with persistently elevated post-intervention levels (1205.31 vs. 1058.02 pg/mg creatinine; $P = 0.007$). Critically, biochemical aspirin resistance incidence was significantly higher in DM patients (20.8%, 11/53) than in controls (7.6%, 5/66; $P = 0.036$) (Table 1).

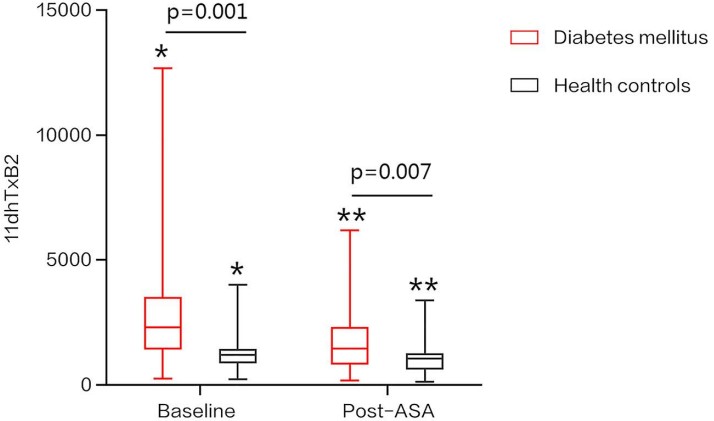

**Fig 1. Urinary 11dhTxB2 levels in DM patients and healthy controls at baseline and post-ASA administration.** Data are presented as median (IQR). *$P = 0.001$, **$P = 0.007$ vs. control group.

**Table 1. Comparison of metabolic, inflammatory, and ASA resistance-related indicators between DM patients and the control group.**

| Variable | DM | Control group | P | U/χ² |
|---|---|---|---|---|
| Age | 62 (57, 71) | 61 (53.75, 70.25) | 0.272 | 1.1 |
| Gender (Male/Female) | 30/23 | 37/29 | 0.953 | 0.004 |
| BMI (kg/m²) | 25.35 (24.30, 26.64) | 24.11 (21.63, 25.93) | 0.003* | 2.970 |
| LDL-C (mmol/L) | 2.76 (2.37, 3.34) | 2.23 (1.63, 2.65) | 0.000* | 4.379 |
| Cysc (mg/L) | 1.24 (0.99, 1.96) | 0.99 (0.84, 1.26) | 0.000* | 3.738 |
| hs-crp(mg/L) | 0.65 (0.44, 0.84) | 0.34 (0.21, 0.46) | 0.000* | 4.185 |
| HOMA-IR | 2.77 (2.03, 4.11) | 1.35 (1.03, 2.26) | 0.000* | 5.285 |
| Pre-ASA 11dhTxB2(pg/mg creatinine) | 2324.58 (1447.50, 3401.08) | 1452.51 (819.70, 2302.12) | 0.001* | 3.310 |
| Post-ASA 11dhTxB2(pg/mg creatinine) | 1205.31 (877.52, 1446.69) | 1058.02 (641.41, 1252.66) | 0.007* | 2.711 |
| Biochemical ASA resistance | 11/53 | 5/66 | 0.036* | 4.387 |

*Note: Comparison between the two groups, P<0.05, indicates a statistically significant difference;BMI: body mass index;11dhTxB2(pg/mg creatinine)= [Urinary 11dhTxB2 (pg/mL)]/ [Urinary creatinine (mg/mL)];Biochemical ASA resistance (poor ASA responder) was defined as post-administration urinary 11dhTxB2 ≥ 1500 pg/mg creatinine.

Fig 2 depicts the urinary 11dhTxB2 level distributions in healthy controls and DM patients at baseline and post-ASA administration. At baseline, healthy controls predominantly exhibited levels below 2000 pg/mg creatinine, whereas DM patients demonstrated a broader distribution range, with a subset exceeding 10,000 pg/mg creatinine. Following ASA administration, both groups showed reduced thromboxane levels, with the control group displaying a tighter post-administration distribution (majority <2000 pg/mg creatinine). Notably, persistent thromboxane overproduction was observed in a proportion of DM patients.

## Univariate analysis of aspirin resistance-related factors in DM patients

To delineate determinants of ASA resistance in DM patients, a comparative analysis was conducted between ASA-resistant and ASA-sensitive subgroups. The results indicated that, compared to the ASA-sensitive group, the ASA-resistant group had significantly higher Pre-11dhTxB2(4719.54 vs 2053.08 pg/mg creatinine, P=0.000), BMI (26.14 vs 25.02 kg/m², P=0.035), HOMA-IR levels (3.51 vs 2.64, P=0.046), and hs-CRP levels (0.75 vs 0.61 mg/L, P=0.032). No significant differences were found between the two groups for other indicators, including GSP, LDL-C, and CysC (P>0.05) (Table 2).

## Multivariable analysis of aspirin resistance-related factors in DM patients

Multivariable logistic regression was employed to identify independent determinants of ASA resistance in DM patients. The results indicated that BMI, GSP, LDL-C, CysC, hs-CRP, and HOMA-IR were not identified as independent factors influencing ASA resistance (all P>0.05) (Table 3).

## Evaluating baseline 11dhTxB2 levels as a predictive biomarker for aspirin resistance in DM patients

Logistic regression analysis further revealed that each unit increase in baseline 11dhTxB2 was associated with a 0.2% increase in the odds of aspirin resistance (OR: 1.002, 95% CI: 1.001–1.003, P=0.002) (Table 4).

## Multiple linear regression analysis of the difference in urinary 11dhTxB2 levels before and after ASA administration in DM patients

Multivariable linear regression identified systemic inflammation (hs-CRP: B=2,147.6, P<0.001) and higher BMI (B=204.9, P=0.021) were independently associated with attenuated thromboxane suppression (Δ11dhTxB2). In contrast, disease duration,GSP, LDL-C,CysC, and HOMA-IR showed no independent associations with Δ11dhTxB2 (all P>0.05) (Table 5).

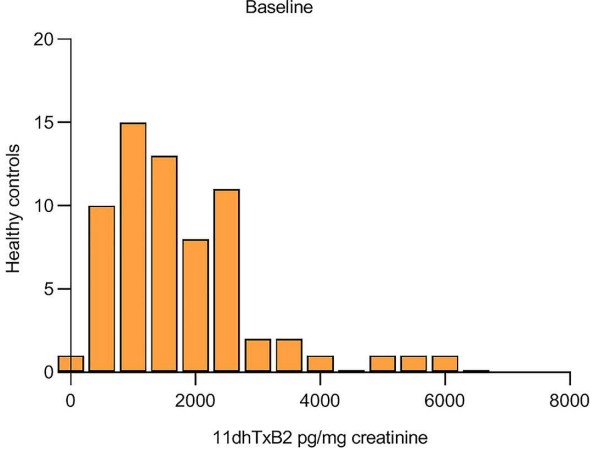

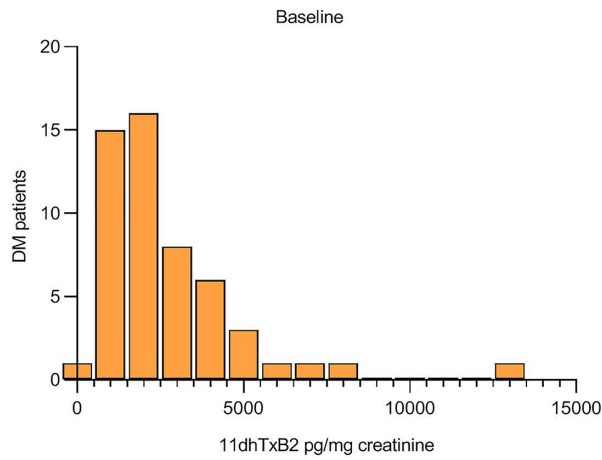

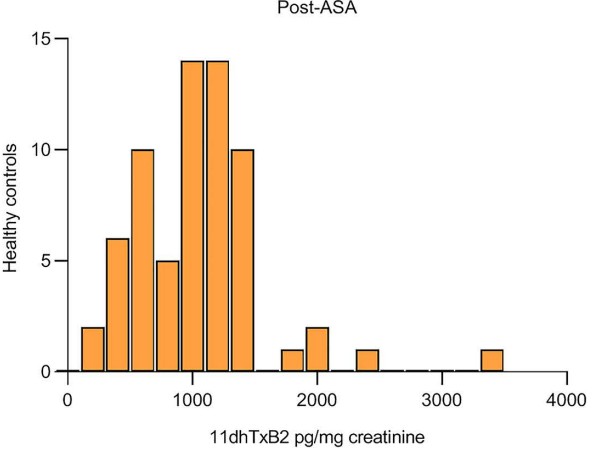

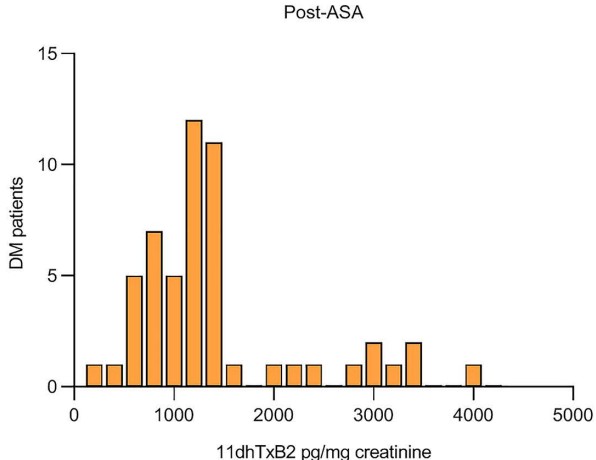

**Fig 2. Frequency distribution of urinary 11dhTxB2 Levels in DM patients and healthy controls at baseline and Post-ASA administration.**

**Table 2. Univariate analysis results of aspirin resistance-related factors in DM.**

| Variable | ASA sensitive | ASA resistance. | P | U |
|---|---|---|---|---|
| Pre-11dhTxB2(pg/mg creatinine) | 2053.08(1314.93, 2461.63) | 4719.54(3765.26, 6270.46) | 0.000 | 440.000 |
| BMI (kg/m²) | 25.02 (24.09, 26.29) | 26.14 (25.86, 27.25) | 0.035 | 2.106 |
| GSP (umol/L) | 281.00 (220.75, 383.75) | 349.00 (310.50, 441.00) | 0.091 | 1.689 |
| LDL-C (mmol/L) | 2.67 (2.36, 3.20) | 3.26 (2.65, 3.47) | 0.151 | 1.437 |
| Cysc (mg/L) | 1.58 (1.01, 1.96) | 1.08 (1.02, 1.61) | 0.476 | −0.713 |
| hs-CRP(mg/L) | 0.61 (0.36, 0.84) | 0.75 (0.73, 0.82) | 0.032 | 2.139 |
| HOMA-IR | 2.64 (2.01, 3.52) | 3.51 (2.96, 4.94) | 0.046 | 1.996 |

**Table 3. Logistic regression analysis results of factors influencing ASA resistance in DM.**

| Influencing factors | β | S.E | Wals | P | OR | OR (95% CI) |
|---|---|---|---|---|---|---|
| BMI | 0.177 | 0.208 | 0.727 | 0.394 | 1.194 | 0.795~1.793 |
| GSP | 0.003 | 0.004 | 0.887 | 0.346 | 1.003 | 0.996~1.010 |
| LDL-C | 0.377 | 0.603 | 0.391 | 0.532 | 1.458 | 0.447~4.748 |
| Cysc | −0.293 | 0.737 | 0.158 | 0.691 | 0.746 | 0.176~3.165 |
| hs-CRP | 0.910 | 1.263 | 0.519 | 0.471 | 2.483 | 0.209~29.505 |
| HOMA-IR | 0.158 | 0.200 | 0.625 | 0.429 | 1.171 | 0.791~1.734 |

**Table 4. Logistic regression analysis of baseline 11dhTxB2 association with biochemical aspirin resistance in DM.**

| Influencing factors | β | S.E | Wals | P | OR | OR (95% CI) |
|---|---|---|---|---|---|---|
| Pre-11dhTxB2 | 0.002 | 0.001 | 9.788 | 0.002 | 1.002 | 1.001~1.003 |

**Table 5. Results of multiple linear regression analysis of the difference in urinary 11dhTxB2 levels before and after ASA administration in DM patients.**

| Parameter | B | SE | Beta | t | P |
|---|---|---|---|---|---|
| hs-CRP | 2147.623 | 517.558 | 0.481 | 4.150 | <0.001 |
| BMI | 204.881 | 86.021 | 0.276 | 2.382 | 0.021 |
| Disease Duration | — | — | 0.203 | 1.543 | 0.129 |
| GSP | — | — | 0.097 | 0.799 | 0.428 |
| LDL-C | — | — | 0.178 | 1.489 | 0.143 |
| Cysc | — | — | 0.058 | 0.463 | 0.646 |
| HOMA-IR | — | — | −0.058 | −0.478 | 0.633 |

## Biochemical aspirin resistance and response variability in DM patients

The Wilcoxon signed-rank test demonstrated a statistically significant decrease in urinary 11dhTxB2 levels following aspirin administration (Z = −6.123, P < 0.001). The median change in 11dhTxB2 levels (Δ11dhTxB2 = pre-aspirin – post-aspirin) was 1119.27 pg/mg creatinine, with an interquartile range (IQR) of 800.00–1500.00 pg/mg creatinine, reflecting considerable inter-individual variability in aspirin response. Of the 53 diabetic patients, 11 (20.8%) exhibited post-aspirin 11dhTxB2 levels ≥1500 pg/mg creatinine and were classified as aspirin-resistant. This subgroup had a median baseline 11dhTxB2 level of 4500 pg/mg creatinine(IQR: 3200–6800 pg/mg creatinine), significantly higher than the overall cohort median of 2324.58 pg/mg creatinine. Notably, a subset of three patients displayed minimal reduction (Δ11dhTxB2 < 10%) despite baseline levels ranging from 1800–2500 pg/mg creatinine.

## Discussion

### The relationship between DM patients and urinary 11dhTxB2 levels

This study revealed significantly elevated baseline urinary 11dhTxB2 levels in DM patients compared to healthy controls (2324.58 vs. 1452.51 pg/mg creatinine; P = 0.001), indicating a prothrombotic phenotype characterized by heightened thromboxane biosynthesis. This finding aligns with established evidence of platelet hyperreactivity in diabetes, characterized by enhanced activation, aggregation, and thromboxane biosynthesis [17,20,21]. Mechanistically, hyperglycemia-induced oxidative stress and endothelial dysfunction amplify COX-1 activity, promoting TxA2 synthesis [22]. Concurrently, insulin resistance—a hallmark of DM—elevates intracellular calcium flux in platelets, potentiating degranulation and

further TxA2 release [23]. Beyond glycemic dysregulation, proatherogenic lipid profiles (e.g., elevated LDL-C) and chronic low-grade inflammation (evidenced by hs-CRP elevation) synergistically enhance platelet adhesion molecule expression (e.g., P-selectin, GPIIb/IIIa), culminating in a self-reinforcing cycle of thromboxane-driven thrombosis [4,24]. Our data thus corroborate the hypothesis that metabolic-inflammatory dysregulation is associated with heightened systemic thromboxane biosynthesis, contributing to increased thrombotic risk in DM.

The elevated baseline urinary 11dhTxB2 levels in diabetic patients (2324.58 vs. 1452.51 pg/mg creatinine in controls, P = 0.001) underscore a heightened thrombotic risk, consistent with prior evidence linking thromboxane biosynthesis to cardiovascular events [16]. Eikelboom et al. (2002) reported that higher 11dhTxB2 levels independently predict myocardial infarction, stroke, or cardiovascular death, with a hazard ratio of 1.8 for the highest versus lowest quartile (P < 0.001) [13]. Post-aspirin, the persistence of elevated 11dhTxB2 (1205.31 vs. 1058.02 pg/mg creatinine in controls, P = 0.007) and a 20.8% prevalence of biochemical resistance (≥1500 pg/mg creatinine) in our diabetic cohort further suggest an attenuated risk reduction. Previous studies indicate that post-ASA 11dhTxB2 levels ≥1500 pg/mg creatinine are associated with a 2- to 4-fold increased risk of adverse cardiovascular outcomes [17], highlighting the potential prognostic utility of this biomarker in identifying high-risk diabetic patients. While our study did not assess clinical events, these findings support the need for personalized risk stratification and therapeutic optimization in this population.

## The effect of aspirin on urinary 11dhTxB2 levels

Following ASA administration, urinary 11dhTxB2 levels decreased by 41.39% in DM patients versus 31.12% in controls, suggesting some suppression of thromboxane biosynthesis despite metabolic dysregulation. Although this numerical disparity favored greater thromboxane suppression in DM, the intergroup difference did not attain statistical significance (P = 0.067), potentially reflecting type II error due to limited sample size or heterogeneous ASA responsiveness within the DM cohort [11,13]. Our observations align with studies demonstrating ASA-mediated attenuation of thromboxane biosynthesis in diabetes [17], yet contrast with reports of "aspirin failure" in subsets with severe insulin resistance or chronic inflammation [12]. Notably, persistent elevation of urinary 11dhTxB2 post-ASA (Fig 2) likely arises predominantly from hyperglycemia-driven COX-2 upregulation in non-platelet cells (e.g., endothelium, monocytes)---a pathway less sensitive to ASA---and/or rapid resynthesis of COX-1 in these cells [12]. Furthermore, glucose variability and oxidative stress in DM may accelerate platelet turnover, replenishing COX-1 pools and blunting ASA's irreversible inhibition [17]. These mechanisms, coupled with interindividual pharmacokinetic differences, likely contribute to the attenuated statistical signal observed.

It is noteworthy that, even in patients classified as ASA-resistant based on the arbitrary cutoff (11dhTxB2 ≥ 1500 pg/mg creatinine), a substantial relative reduction in 11dhTxB2 levels—such as the 41.39% decrease observed in the DM group—may still indicate clinical benefit. Previous studies have demonstrated that urinary 11dhTxB2 levels correlate with cardiovascular outcomes [13], suggesting that any reduction in thromboxane biosynthesis could mitigate thrombotic risk. Thus, the clinical implications of ASA resistance should be interpreted cautiously.

## Exploration of the mechanisms of aspirin resistance

This study further examined the factors related to aspirin resistance in diabetic patients. Univariate analysis revealed that BMI, HOMA-IR, and hs-CRP were significantly associated with ASA resistance, consistent with the findings of several studies. Previous studies have indicated a strong association between higher BMI and ASA resistance. Overweight or obese individuals often exhibit higher levels of inflammation and abnormal blood glucose, which may be linked to their poor response to ASA [25,26]. Additionally, studies have shown that hs-CRP, a blood marker of inflammation, is closely associated with the antithrombotic effect of aspirin [27]. Our results further support these findings, suggesting that chronic inflammatory states may reduce aspirin efficacy by enhancing platelet activation or altering thrombosis formation mechanisms.

The discrepancy between univariate and multivariable analyses may reflect collinearity among metabolic-inflammatory variables (e.g., BMI and hs-CRP) or heterogeneous ASA responsiveness within the DM cohort. For instance, subsets with severe insulin resistance or genetic polymorphisms in COX-1/PEAR1 may exhibit distinct pharmacodynamic profiles, diluting the independent effect of individual factors in logistic regression. This result suggests that these variables may not be independent factors influencing aspirin resistance. The mechanisms of aspirin resistance are complex, and factors such as genetics, changes in platelet function, aspirin metabolism, small sample sizes, or other confounding factors may interact to influence the results of this study [28,29].

Furthermore, our study identified baseline 11dhTxB2 levels as a significant predictor of aspirin resistance in diabetic patients. The univariate analysis showed markedly higher baseline levels in the ASA-resistant subgroup (4719.54 vs. 2053.08 pg/mg creatinine, P = 0.000), and logistic regression confirmed that each unit increase in baseline 11dhTxB2 was associated with a 0.2% higher odds of resistance (OR: 1.002, 95% CI: 1.001–1.003, P = 0.002). This suggests that elevated thromboxane biosynthesis at baseline may reflect a prothrombotic state less responsive to ASA, potentially driven by accelerated platelet turnover or non-COX-1 pathways, predominantly originating from non-platelet sources as discussed, warranting further mechanistic exploration.

Emerging evidence suggests that non-platelet sources of thromboxane (e.g., endothelial COX-2) may contribute to residual thromboxane biosynthesis despite ASA therapy [17]. Furthermore, accelerated platelet turnover in diabetic patients could replenish uninhibited cyclooxygenase-1 pools, potentially explaining the lack of independent associations in multivariable models [25].

### The role of oxidative stress in thromboxane biosynthesis

A critical consideration not addressed in our analysis is the potential contribution of oxidative stress to persistent thromboxane generation. Hyperglycemia in diabetes potently induces oxidative stress through mitochondrial superoxide overproduction, advanced glycation end-product (AGE) formation, and protein kinase C (PKC) activation [30]. This pro-oxidant state directly upregulates COX-2 expression in vascular endothelial cells and monocytes [31], while simultaneously enhancing substrate availability for thromboxane synthesis via lipid peroxidation [32,33]. Compelling evidence positions oxidative stress as a primary driver of systemic thromboxane generation, potentially surpassing traditional risk factors. In the large-scale Framingham Heart Study cohort, oxidative stress markers demonstrated the strongest association with urinary 11dhTxB2 levels among over 3,000 participants, exceeding the predictive value of diabetes status or systemic inflammation [34]. This suggests oxidative stress may be a fundamental mechanism underlying the attenuated aspirin response observed in our diabetic cohort, particularly through its stimulation of non-platelet thromboxane pathways.

### The role of hs-CRP and BMI in ASA response

Multiple linear regression analysis showed that hs-CRP, a marker of systemic inflammation, and BMI were significantly associated with the difference in urinary 11dhTxB2 levels after ASA administration in diabetic patients. Previous studies have indicated that hs-CRP, is closely associated with the risk of cardiovascular diseases [35,36]. Additionally, the relationship between BMI, cardiovascular events, and drug responsiveness has been extensively studied [37,38]. Higher BMI not only contributes to insulin resistance and chronic low-grade inflammation, but it may also directly affect platelet function and thrombus formation [39,40]. The results of this study further validate these findings, suggesting that the attenuation of total thromboxane biosynthesis suppression by aspirin in diabetic patients may be related to the combined presence of these factors, potentially through their promotion of non-platelet (e.g., COX-2 mediated) TxA2 production.

Persistent thromboxane overproduction post-ASA may arise from hyperglycemia-driven COX-2 upregulation in endothelial cells or monocytes—a pathway less sensitive to ASA inhibition [31]. Emerging evidence highlights that adipose tissue-derived exosomes in obese individuals carry microRNAs (e.g., miR-155) capable of enhancing platelet COX-1

expression, potentiating thromboxane biosynthesis despite ASA therapy [41]. Furthermore, systemic inflammation in diabetes may induce COX-2-mediated thromboxane production in endothelial cells, a pathway resistant to ASA inhibition [42].

## Heterogeneity in aspirin response among patients with diabetes

Despite a greater median reduction in 11dhTxB2 levels in the DM group (1119.27 pg/mg creatinine) compared to controls (394.49 pg/mg creatinine), fewer DM patients achieved the threshold of <1500 pg/mg (79.2% vs. 92.4%). This is likely due to significantly higher baseline 11dhTxB2 levels in the DM group (median: 2324.58 pg/mg creatinine) requiring a larger absolute reduction to fall below the threshold.

Our findings highlight heterogeneity in aspirin response among diabetic patients, with a small subgroup showing minimal reduction in 11dhTxB2 levels despite moderate baseline values. This observation points to the existence of true aspirin resistance, potentially linked to mechanisms such as accelerated platelet turnover or, more likely given the biomarker used, robust non-platelet (e.g., COX-2 mediated) thromboxane production, consistent with previous reports [6,17]. These results emphasize the importance of tailoring aspirin therapy, particularly for high-risk diabetic patients with elevated baseline thromboxane levels or suboptimal responses to standard doses.

## Limitations

Several limitations should be acknowledged. First, the cohort's modest sample size (53 diabetic patients and 66 controls) may restrict the generalizability of our findings, particularly given the clinical and metabolic heterogeneity inherent to diabetic populations. A larger, multicenter cohort with predefined stratification criteria could enhance external validity. Second, the cross-sectional design precludes causal inference—while metabolic-inflammatory markers (BMI, hs-CRP, HOMA-IR) correlate with aspirin resistance, their temporal relationship and mechanistic contributions remain undefined. The observed associations between metabolic-inflammatory markers (e.g., BMI, hs-CRP) and attenuated aspirin response (Δ11dhTxB2 or ASA resistance status) should therefore be interpreted as correlational, not causal. Longitudinal studies tracking dynamic changes in platelet reactivity alongside metabolic parameters are needed to establish causality.

Third, although urinary 11dhTxB2 is a validated biomarker of thromboxane biosynthesis, notably, complementary assessments of platelet function—such as agonist-induced aggregation assays or flow cytometric analysis of activation markers—were not performed. Moreover, unmeasured inflammatory mediators (e.g., IL-6, TNF-α) and oxidative stress markers (e.g., 8-iso-prostaglandin F2α, malondialdehyde) could provide further mechanistic insights into COX-2-driven thromboxane biosynthesis. Oxidative stress is a particularly significant unmeasured confounder, as it has been robustly established as one of the strongest predictors of thromboxane generation—even surpassing traditional risk factors like diabetes and inflammation in large cohort studies [32–34]. In aspirin-treated individuals, oxidative stress predominantly drives non-platelet (e.g., endothelial/monocytic) TxA2 production via COX-2 upregulation, directly contributing to thrombosis risk and mortality [32,33]. Our observed associations between metabolic-inflammatory markers (hs-CRP, HOMA-IR) and attenuated thromboxane suppression may partly reflect underlying oxidative stress pathways.

Fourth, our reliance on urinary 11dhTxB2 as the sole measure of aspirin response is an important limitation. As discussed, this biomarker reflects total body TxA2 biosynthesis, and post-aspirin levels are predominantly influenced by non-platelet sources. We did not employ platelet-specific functional assays (e.g., light transmission aggregometry in response to arachidonic acid or collagen, VerifyNow Aspirin test, or thromboxane-dependent platelet activation markers like P-selectin expression) which could more directly assess the inhibition of platelet COX-1-dependent pathways. Consequently, our definition of "biochemical resistance" captures systemic thromboxane overproduction under therapy but cannot definitively distinguish true platelet COX-1 resistance from significant non-platelet contributions. Future studies incorporating both urinary metabolites and platelet-specific functional assays would provide a more comprehensive picture.

Regarding methodology, the use of an arbitrary cutoff for ASA resistance (urinary 11dhTxB2 ≥ 1500 pg/mg creatinine) may not fully reflect the clinical benefit of aspirin, particularly in patients who exhibit substantial relative reductions in 11dhTxB2 levels. To address this limitation, future studies should consider both absolute and relative changes in 11dhTxB2 to better elucidate the relationship between biochemical markers and clinical outcomes.

## Conclusion

In summary, this study demonstrates that patients with diabetes exhibit a prothrombotic phenotype characterized by elevated baseline urinary 11dhTxB2 levels, which also serve as a predictive biomarker for biochemical aspirin resistance (OR: 1.002, P = 0.002). The incidence of biochemical aspirin resistance was 2.7-fold higher in diabetic patients compared to healthy controls (20.8% vs. 7.6%, P = 0.036). Notably, metabolic-inflammatory dysregulation—reflected by higher BMI and hs-CRP—emerged as key factors associated with attenuated aspirin efficacy, with persistent thromboxane biosynthesis (likely originating predominantly from non-platelet sources) post-intervention suggesting pathways beyond conventional platelet COX-1 inhibition, such as COX-2 in nucleated cells or rapidly regenerated COX-1 in diabetic thrombogenesis.

While ASA intervention reduced urinary 11dhTxB2 by 41.4% in diabetic patients versus 31.1% in controls, the intergroup difference (P = 0.067) and residual thromboxane overproduction highlight the unmet need for personalized antiplatelet strategies targeting platelet hyperreactivity driven by insulin resistance and chronic inflammation. These findings underscore the imperative to integrate metabolic control with tailored antiplatelet therapies in diabetes management.

Future investigations should prioritize:

1. Mechanistic studies dissecting platelet turnover dynamics and non-platelet thromboxane sources (e.g., endothelial COX-2) in diabetic populations,incorporating direct measurements of oxidative stress biomarkers (e.g., urinary 8-iso-PGF2α, plasma oxidized LDL);

2. Pharmacogenomic profiling to identify genetic variants(e.g.,COX-1 haplotypes, PEAR1 polymorphisms) influencing ASA pharmacodynamics;

3. Randomized trials evaluating combinatorial therapies targeting oxidative stress pathways (e.g., ASA + NRF2 activators like sulforaphane or mitochondria-targeted antioxidants) or anti-inflammatory agents (e.g., IL-6 inhibitors) to mitigate inflammation-enhanced thrombotic risk.

## Supporting information

**S1 File. Data.**
(XLSX)

## Acknowledgments

We express our sincere gratitude to all diabetic patients and healthy volunteers who participated in this study. We are deeply grateful to the healthcare professionals from the Department of Endocrinology and nuclear medicine technicians at Chaohu Hospital Affiliated to Anhui Medical University for their dedicated support and technical assistance.

## Author contributions

**Conceptualization:** Bo Chen, Zisheng Li.

**Data curation:** Bo Chen, Zisheng Li.

**Investigation:** Jianyong Zhao, Huamei Dong, Long Tong, Jiaqing Dou.

**Writing – original draft:** Bo Chen.

**Writing – review & editing:** Bo Chen.

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
