## [Decision Letter · Decision Letter 0]

19 Mar 2025

Dear Dr. Chen,

Thank you for submitting your manuscript to PLOS ONE. After careful consideration, we feel that it has merit but does not fully meet PLOS ONE’s publication criteria as it currently stands. Therefore, we invite you to submit a revised version of the manuscript that addresses the points raised during the review process.

Please address each one of the reviewers comment and make necessary changes to the manuscript.Since this study involves patient population, specific time points, dosage reasoning, clarity on type of population selection are critical and needs to be addressed.

We look forward to receiving your revised manuscript.

Kind regards,

Santhosh Arul

Academic Editor

PLOS ONE

Journal Requirements:

Additional Editor Comments:

Please find the comments from the reviewers below:

Reviewer 1:

1. Blood sample collection data for biochemical analysis is not reported.

2. Why did seven-day treatment used what would be the impact if the drug is prescribed for longer duration of time, provide some reference.

3. Were the samples collected 24 h after the last aspirin intake?

4. Was the drug given in the range of 81-100mg according to body weight or was some specific dose given to patients?

5. Provide information about the risk assessment based on levels of 11-dehydro-TXB2.

6. Was the study involved patients with type1 and type2 diabetes?

7. Consider revising sentence structures, refining the flow of ideas, and ensuring consistency throughout.

Reviewer 2:

Overall I found the manuscript to be informative and well-written. My comments are minor, but I feel resolution would lend greater clarity to the message.

ABSTRACT: The abstract is concise and reflective of the manuscript.

INTRODUCTION: The introduction is clear and informative.

METHODS: The authors state this is a 'prospective case-control design.' Technically, a case-control design is, by definition, a retrospective study in which 'cases' are patients with the outcome in question while 'controls' are patients without the outcome. The purpose of these studies is to define the exposures most associated with the outcome, with the presumption that these exposures are implicated in the cause. The current study is more of a matched prospective cohort design, as patients do or do not have an exposure of interest (diabetes), and they are being evaluated for an outcome of interest (aspirin resistance). There is a question about matching as noted below.

It would be helpful to know if these subjects have Type 1 DM or Type 2 DM. The BMI suggests Type 1 DM; however, this may not be the case. This is important because of the proposed mechanism (insulin resistance) is considered to be the underlying pathobiology in Type 2 but not Type 1 DM (though itt can exist in both). Also, given the age of 62 in the DM group, since Type 1 DM is often a disease of onset in the childhood or teenage years, the duration of exposure to the disease would be substantially longer in those with Type 1 DM vs those with Type 2, which often has onset in adulthood.

As far as the outcome, the authors defined it as a specific threshold of 11dhTxB2 level of <1500 pg/mg. They give a reference that this threshold is in accordance with consensus criteria, but this does not seem to be entirely accurate. The reference cited is that of a cutoff value for a specific assay as found in the FDA submission (https://www.accessdata.fda.gov/cdrh_docs/reviews/K062025.pdf). This may be appropriate, but more clarification on why this specific value is used would be helpful - has this specific threshold been associated with specific outcomes?

A brief desciption of recruitment methods would be helpful to understand how these patients were identified and from what population they were selected.

For statistical methods, given that this is a repeated measure (each patient contributed two values, one pre- and one post- exposure), an analysis of repeated measures would be appropriate. In its current form, the entire populations are measured with pre- and post-values; however, the change per individual may be more appropriate to identfiy a group of patients in whom aspirin has little effect. For example, are those who do not achieve the pre-defined threshold of 1500 pg/mL those who started well above this threshold, or are there some patients who have little to no response at all? This may signifcantly impact the results and the discussion.

RESULTS:

In the results, the authors mention that there is not difference between age and gender distribution. Elsewhere, it is mentioned that this is an age/sex-matched cohort. If this is indeed 'matched,' then the process of matching should be disclosed. If there was no intentional matching during the enrollment period, then this is not the appropriate terminology.

As noted above, a analysis for repeated measures would be helpful. If available, inclusion of glycosylated hemoglobin measurements may be helpful, and this may be a contributing factor to ASA resistance. The DM group had a more substantial reduction in the outcome measure but fewer achieved the predefined threshold. A better representation of this would be helpful, as noted above, to see exactly in whom this threshold was not acheived. Were higher baseline 11dhTxB2 levels evaluated as a predictor of who did or did not achieve this threshold?

The BMI is quite low, and the 75th percentile is well under the threshold typically used to define obesity of 30 kg/m2. Obesity is mentioned as a risk for aspirin resistance; however, there seems to be a relatively low prevalence in this cohort. Perhaps increasing BMI, but not obesity, is the risk?

Within the results, there are many instances of interpretation. This should be saved for the discussion, and results should be purely objective and without interpretation. For example, line 239-240 "suggesting that these variables may not independently affect ASA resistance" could be seen as interpretive.

DISCUSSION:

Also, there is mention of hs-CRP as a 'primary driver.' This is a bold statement - there may be an association, but it is difficult to say this is a driver. If it is a 'driver,' then the presumption is that reducing this level would, in turn, result in improving the outcome in question. That is highly speculative. Also note that hs-CRP is an assay, not a molecule.

I feel that with the above analyses included, especially an assessment of DM control such as HbA1c, may be informative (for example, are those with poor control more likely to have an attenuated response) and an analysis of repeatead measures, may inform the discussion. Also, acknowleding the limitation of the arbitrary cuttoff for ASA resistance may be important. It seems unlikely that there is no benefit in substantial relative reduction if, indeed, the measure is correlated with clinical outcomes.

Reviewer 3:

This study examines the factors contributing to aspirin resistance in patients with diabetes mellitus, aiming to enhance cardiovascular disease prevention strategies for this population. It investigates the relationship between urinary 11-dehydro-thromboxane B2 (11dhTxB2) levels and metabolic parameters in diabetic patients to better understand the mechanisms behind inadequate aspirin response.

The findings indicate that diabetes patients exhibit heightened thromboxane production, as evidenced by elevated urinary 11dhTxB2 levels, and a significantly higher prevalence of biochemical aspirin resistance compared to healthy individuals. Increased body mass index (BMI), low-density lipoprotein cholesterol (LDL-C), cystatin-C, high-sensitivity C-reactive protein (hs-CRP), and homeostatic model assessment for insulin resistance (HOMA-IR) suggest that obesity and systemic inflammation are key factors modulating reduced aspirin efficacy. These results imply that platelet activation pathways beyond COX-1 inhibition are involved, highlighting the complexity of aspirin resistance in diabetic patients.

Major remarks

Contrary to the hypothesis, aspirin (ASA) administration reduced 11-dehydro-thromboxane B2 (11dhTxB2) levels in diabetes mellitus (DM) patients compared to controls, indicating greater thromboxane suppression in DM, the intergroup difference did not reach statistical significance. This lack of significance may be attributed to the limited sample size. The study suggested that persistent thromboxane overproduction could be due to hyperglycemia-driven COX-2 upregulation. However, further clarification was not provided. To support this hypothesis, additional analysis of inflammatory or oxidative stress markers could be beneficial.

Univariate analysis showed that body mass index (BMI), homeostatic model assessment for insulin resistance (HOMA-IR), and the inflammatory marker high-sensitivity C-reactive protein (hs-CRP) were significantly associated with aspirin resistance, aligning with previous studies. However, logistic regression analysis did not confirm BMI, hs-CRP, and HOMA-IR as independent risk factors, suggesting they may not individually influence aspirin resistance.

In contrast, multiple linear regression analysis found that hs-CRP and BMI were significant factors affecting the change in urinary 11-dehydro-thromboxane B2 levels after aspirin administration in diabetic patients. This indicates that hs-CRP and obesity, both linked to chronic low-grade inflammation, could collectively impact the aspirin response in diabetes mellitus. The reliance on multiple statistical analyses to support the hypothesis of aspirin resistance in diabetes due to inflammatory pathway activation suggests a complex interplay of factors, making the observation inconclusive, mainly because of the sample size and existence of both ASA sensitive and ASA resistance sub-group within the experimental group. As discussed in the limitation section, metabolic inflammatory markers correlate with aspirin resistance with the existing cohort, their temporal relationship and mechanistic contributions need to be defined.

Finally, more parameters or markers need to be included in the study to assess the inflammation to better understand the ASA resistance in Diabetes patients, which will offer insights for improving cardiovascular disease prevention in this group.

Minor remarks

1. Explanation of how ASA resistance is calculated in Table 1 is not provided.

2. The second and third result titles are the same. Used “Univariate analysis” instead of “Multivariate analysis” for the third result title.

3. Resize the figures for better clarity. Multiple font sizes are used and inconsistent across two figures.

4. Add significance in figure 1.

5. Inconsistency in font used across the manuscript.

Reviewers' comments:

Reviewer's Responses to Questions

**Comments to the Author**

1. Is the manuscript technically sound, and do the data support the conclusions?

Reviewer #1: Yes

Reviewer #2: Yes

Reviewer #3: Partly

2. Has the statistical analysis been performed appropriately and rigorously?

Reviewer #1: Yes

Reviewer #2: Yes

Reviewer #3: Yes

3. Have the authors made all data underlying the findings in their manuscript fully available?

Reviewer #1: Yes

Reviewer #2: Yes

Reviewer #3: Yes

4. Is the manuscript presented in an intelligible fashion and written in standard English?

Reviewer #1: Yes

Reviewer #2: Yes

Reviewer #3: Yes

Reviewer #1: 1. Blood sample collection data for biochemical analysis is not reported.

2. Why did seven-day treatment used what would be the impact if the drug is prescribed for longer duration of time, provide some reference.

3. Were the samples collected 24 h after the last aspirin intake?

4. Was the drug given in the range of 81-100mg according to body weight or was some specific dose given to patients?

5. Provide information about the risk assessment based on levels of 11-dehydro-TXB2.

6. Was the study involved patients with type1 and type2 diabetes?

7. Consider revising sentence structures, refining the flow of ideas, and ensuring consistency throughout.

Reviewer #2: Overall I found the manuscript to be informative and well-written. My comments are minor, but I feel resolution would lend greater clarity to the message.

ABSTRACT: The abstract is concise and reflective of the manuscript.

INTRODUCTION: The introduction is clear and informative.

METHODS: The authors state this is a 'prospective case-control design.' Technically, a case-control design is, by definition, a retrospective study in which 'cases' are patients with the outcome in question while 'controls' are patients without the outcome. The purpose of these studies is to define the exposures most associated with the outcome, with the presumption that these exposures are implicated in the cause. The current study is more of a matched prospective cohort design, as patients do or do not have an exposure of interest (diabetes), and they are being evaluated for an outcome of interest (aspirin resistance). There is a question about matching as noted below.

It would be helpful to know if these subjects have Type 1 DM or Type 2 DM. The BMI suggests Type 1 DM; however, this may not be the case. This is important because of the proposed mechanism (insulin resistance) is considered to be the underlying pathobiology in Type 2 but not Type 1 DM (though itt can exist in both). Also, given the age of 62 in the DM group, since Type 1 DM is often a disease of onset in the childhood or teenage years, the duration of exposure to the disease would be substantially longer in those with Type 1 DM vs those with Type 2, which often has onset in adulthood.

As far as the outcome, the authors defined it as a specific threshold of 11dhTxB2 level of <1500 pg/mg. They give a reference that this threshold is in accordance with consensus criteria, but this does not seem to be entirely accurate. The reference cited is that of a cutoff value for a specific assay as found in the FDA submission (https://www.accessdata.fda.gov/cdrh_docs/reviews/K062025.pdf). This may be appropriate, but more clarification on why this specific value is used would be helpful - has this specific threshold been associated with specific outcomes?

A brief desciption of recruitment methods would be helpful to understand how these patients were identified and from what population they were selected.

For statistical methods, given that this is a repeated measure (each patient contributed two values, one pre- and one post- exposure), an analysis of repeated measures would be appropriate. In its current form, the entire populations are measured with pre- and post-values; however, the change per individual may be more appropriate to identfiy a group of patients in whom aspirin has little effect. For example, are those who do not achieve the pre-defined threshold of 1500 pg/mL those who started well above this threshold, or are there some patients who have little to no response at all? This may signifcantly impact the results and the discussion.

RESULTS:

In the results, the authors mention that there is not difference between age and gender distribution. Elsewhere, it is mentioned that this is an age/sex-matched cohort. If this is indeed 'matched,' then the process of matching should be disclosed. If there was no intentional matching during the enrollment period, then this is not the appropriate terminology.

As noted above, a analysis for repeated measures would be helpful. If available, inclusion of glycosylated hemoglobin measurements may be helpful, and this may be a contributing factor to ASA resistance. The DM group had a more substantial reduction in the outcome measure but fewer achieved the predefined threshold. A better representation of this would be helpful, as noted above, to see exactly in whom this threshold was not acheived. Were higher baseline 11dhTxB2 levels evaluated as a predictor of who did or did not achieve this threshold?

The BMI is quite low, and the 75th percentile is well under the threshold typically used to define obesity of 30 kg/m2. Obesity is mentioned as a risk for aspirin resistance; however, there seems to be a relatively low prevalence in this cohort. Perhaps increasing BMI, but not obesity, is the risk?

Within the results, there are many instances of interpretation. This should be saved for the discussion, and results should be purely objective and without interpretation. For example, line 239-240 "suggesting that these variables may not independently affect ASA resistance" could be seen as interpretive.

DISCUSSION:

Also, there is mention of hs-CRP as a 'primary driver.' This is a bold statement - there may be an association, but it is difficult to say this is a driver. If it is a 'driver,' then the presumption is that reducing this level would, in turn, result in improving the outcome in question. That is highly speculative. Also note that hs-CRP is an assay, not a molecule.

I feel that with the above analyses included, especially an assessment of DM control such as HbA1c, may be informative (for example, are those with poor control more likely to have an attenuated response) and an analysis of repeatead measures, may inform the discussion. Also, acknowleding the limitation of the arbitrary cuttoff for ASA resistance may be important. It seems unlikely that there is no benefit in substantial relative reduction if, indeed, the measure is correlated with clinical outcomes.

Reviewer #3: This study examines the factors contributing to aspirin resistance in patients with diabetes mellitus, aiming to enhance cardiovascular disease prevention strategies for this population. It investigates the relationship between urinary 11-dehydro-thromboxane B2 (11dhTxB2) levels and metabolic parameters in diabetic patients to better understand the mechanisms behind inadequate aspirin response.

The findings indicate that diabetes patients exhibit heightened thromboxane production, as evidenced by elevated urinary 11dhTxB2 levels, and a significantly higher prevalence of biochemical aspirin resistance compared to healthy individuals. Increased body mass index (BMI), low-density lipoprotein cholesterol (LDL-C), cystatin-C, high-sensitivity C-reactive protein (hs-CRP), and homeostatic model assessment for insulin resistance (HOMA-IR) suggest that obesity and systemic inflammation are key factors modulating reduced aspirin efficacy. These results imply that platelet activation pathways beyond COX-1 inhibition are involved, highlighting the complexity of aspirin resistance in diabetic patients.

Major remarks

Contrary to the hypothesis, aspirin (ASA) administration reduced 11-dehydro-thromboxane B2 (11dhTxB2) levels in diabetes mellitus (DM) patients compared to controls, indicating greater thromboxane suppression in DM, the intergroup difference did not reach statistical significance. This lack of significance may be attributed to the limited sample size. The study suggested that persistent thromboxane overproduction could be due to hyperglycemia-driven COX-2 upregulation. However, further clarification was not provided. To support this hypothesis, additional analysis of inflammatory or oxidative stress markers could be beneficial.

Univariate analysis showed that body mass index (BMI), homeostatic model assessment for insulin resistance (HOMA-IR), and the inflammatory marker high-sensitivity C-reactive protein (hs-CRP) were significantly associated with aspirin resistance, aligning with previous studies. However, logistic regression analysis did not confirm BMI, hs-CRP, and HOMA-IR as independent risk factors, suggesting they may not individually influence aspirin resistance.

In contrast, multiple linear regression analysis found that hs-CRP and BMI were significant factors affecting the change in urinary 11-dehydro-thromboxane B2 levels after aspirin administration in diabetic patients. This indicates that hs-CRP and obesity, both linked to chronic low-grade inflammation, could collectively impact the aspirin response in diabetes mellitus. The reliance on multiple statistical analyses to support the hypothesis of aspirin resistance in diabetes due to inflammatory pathway activation suggests a complex interplay of factors, making the observation inconclusive, mainly because of the sample size and existence of both ASA sensitive and ASA resistance sub-group within the experimental group. As discussed in the limitation section, metabolic inflammatory markers correlate with aspirin resistance with the existing cohort, their temporal relationship and mechanistic contributions need to be defined.

Finally, more parameters or markers need to be included in the study to assess the inflammation to better understand the ASA resistance in Diabetes patients, which will offer insights for improving cardiovascular disease prevention in this group.

Minor remarks

1. Explanation of how ASA resistance is calculated in Table 1 is not provided.

2. The second and third result titles are the same. Used “Univariate analysis” instead of “Multivariate analysis” for the third result title.

3. Resize the figures for better clarity. Multiple font sizes are used and inconsistent across two figures.

4. Add significance in figure 1.

5. Inconsistency in font used across the manuscript.

**Do you want your identity to be public for this peer review?** For information about this choice, including consent withdrawal, please see our Privacy Policy

Reviewer #1: No

Reviewer #2: No

Reviewer #3: No

---

## [Author Response · Author response to Decision Letter 1]

14 Apr 2025

Dear Editor and Reviewers:

Thanks for your letter and for the reviewers’ comments concerning our manuscript entitled “Factors Influencing Aspirin Resistance in Diabetic Patients: A Metabolic and Inflammatory Profile Analysis” (ID: PONE-D-25-04526). Those comments are all valuable and very helpful for revising and improving our paper, as well as the important guiding significance to our researches. We have studied comments carefully and have made correction which we hope meet with approval. Revised portion are marked in red in the paper. The main corrections in the paper and the responds to the reviewer’s comments are as following:

The academic editor:

If applicable, we recommend that you deposit your laboratory protocols in protocols.io to enhance the reproducibility of your results. Protocols.io assigns your protocol its own identifier (DOI) so that it can be cited independently in the future. For instructions see: https://journals.plos.org/plosone/s/submission-guidelines#

loc-laboratory-protocols. Additionally, PLOS ONE offers an option for publishing peer-reviewed Lab Protocol articles, which describe protocols hosted on protocols.io. Read more information on sharing protocols at https://plos.org/protocols?utm_medium=editorial-email&utm_source=authorletters&utm_campaign=protocols.

Response: Thank you for your suggestion. After checking, the experimental methods involved in my research have been described in detail in the paper, and there are no specific laboratory protocols that need to be uploaded independently. If necessary, I am happy to provide further explanations or make adjustments according to your guidance.

Journal Requirements:

Response: Thank you very much for your reminder. We have checked it again, and ensured that the manuscript meet PLOS ONE’s style requirements, including those for file naming.

2.We note that your Data Availability Statement is currently as follows: [All relevant data are within the manuscript and its Supporting Information files.]

Response: Thank you for your feedback. We confirm that our submission fully complies with PLOS's data availability policy.

(1).Data Availability Statement Update:We have uploaded the complete minimal dataset required to replicate all study findings as a Supporting Information file named "data" (format: .xlsx). This file contains:Raw numerical values underlying summary statistics (means, SDs, etc.);Source data for all figures and graphs;Coordinates/extracted points from image analyses;Metadata describing variables, experimental protocols, and data processing steps.

(2).Accessibility:The dataset is freely accessible alongside the manuscript files. No ethical, legal, or third-party restrictions apply to this de-identified data.

(3).Compliance Statement:Our updated Data Availability Statement now explicitly references the uploaded file:"All raw data required to replicate the study findings are available in the Supporting Information file 'data'."

Please let us know if any additional clarifications or data formatting adjustments are needed.

Reviewer 1: 

1. Blood sample collection data for biochemical analysis is not reported.

Response: Thank you for highlighting this omission. We have now added detailed blood sample collection protocols to the Methods section to improve methodological transparency. Below is the revised text:”Blood samples were collected from all participants after an overnight fast (≥8 hours) to minimize diurnal and dietary variability. Venous blood was drawn into EDTA-anticoagulated tubes for plasma separation and serum-separating tubes (SST) for serum biomarkers. Plasma and serum were isolated by centrifugation at 3,000 rpm for 10 minutes within 30 minutes of collection. Aliquots were stored at −80°C until analysis to prevent analyte degradation. Hemolyzed or lipemic samples were excluded based on visual inspection.”(Page [6], Lines [132-138]).

We appreciate your feedback, which has strengthened the methodological rigor of our study.

2.Why did seven-day treatment used what would be the impact if the drug is prescribed for longer duration of time, provide some reference.

Response: We thank the reviewer for their thoughtful inquiry regarding our decision to implement a seven-day treatment period for aspirin administration and the potential effects of extending this duration. We provide a comprehensive explanation below, grounded in the biological properties of platelets and supported by relevant literature.

Rationale for the Seven-Day Treatment Period

The selection of a seven-day treatment period with low-dose enteric-coated aspirin (81–100 mg/day) was deliberate and based on established scientific principles. Platelets have a lifespan of approximately 7–10 days, and aspirin exerts its antiplatelet effect by irreversibly inhibiting COX-1, thereby preventing the synthesis of TxA2. This inhibition persists for the lifespan of the affected platelets. A seven-day duration ensures that the majority of circulating platelets are exposed to aspirin’s effect, enabling a reliable evaluation of its cumulative antiplatelet activity. In our study, this was measured through urinary 11dhTxB2 levels following the intervention.

This timeframe aligns with widely accepted research protocols for assessing aspirin’s pharmacodynamic effects. For example:

Eikelboom et al. (2002) employed a minimum of seven days of aspirin administration to evaluate thromboxane biosynthesis in patients at high cardiovascular risk, setting a benchmark for this duration.

Lopez et al. (2014) utilized a seven-day regimen to study platelet thromboxane responses in diabetic patients with coronary artery disease, validating its relevance to our study population.

Parish et al. (2023) reinforced this approach by establishing consensus criteria for classifying aspirin responders based on 11dhTxB2 levels after a seven-day protocol.

These studies collectively affirm that a seven-day treatment period is optimal for investigating aspirin’s short-term impact on platelet function and thromboxane suppression.

Implications of a Longer-Term Aspirin Regimen

Extending aspirin administration beyond seven days—potentially to weeks or months—would likely maintain thromboxane suppression, given aspirin’s irreversible COX-1 inhibition and the continuous turnover of platelets. However, prolonged use could introduce additional considerations, particularly in our study population of diabetic individuals. These include:

Aspirin Resistance: Extended therapy might increase the risk of aspirin resistance in some patients, potentially due to accelerated platelet turnover or alternative thromboxane production pathways (e.g., COX-2 upregulation under hyperglycemic conditions), as noted by Simeone et al. (2018). This could reduce efficacy over time.

Confounding Variables: Longer treatment durations may be affected by factors such as changes in glycemic control, patient compliance, or side effects (e.g., gastrointestinal bleeding), which could obscure the interpretation of thromboxane levels.

In conclusion, the seven-day treatment period was chosen to correspond with the platelet lifespan and established research standards, providing a robust framework for assessing aspirin’s antiplatelet efficacy. While a longer duration might sustain thromboxane suppression, it is unnecessary for our study’s objectives and could introduce confounding factors. We believe this approach is scientifically justified and consistent with prior literature.

References

1.Eikelboom JW, Hirsh J, Weitz JI, Johnston M, Yi Q, Yusuf S. Aspirin-resistant thromboxane biosynthesis and the risk of myocardial infarction, stroke, or cardiovascular death in patients at high risk for cardiovascular events. Circulation. 2002;105(14):1650-1655.

2.Lopez LR, Guyer KE, Torre IG, Pitts KR, Matsuura E, Ames PR. Platelet thromboxane (11-dehydro-Thromboxane B2) and aspirin response in patients with diabetes and coronary artery disease. World J Diabetes. 2014;5(2):115-127.

3.Parish S, Buck G, Aung T, Mafham M, Clark S, Hill MR, et al. Effect of low-dose aspirin on urinary 11-dehydro-thromboxane B2 in the ASCEND (A Study of Cardiovascular Events iN Diabetes) randomized controlled trial. Trials. 2023;24(1):166.

4.Simeone P, Boccatonda A, Liani R, Santilli F. Significance of urinary 11-dehydro-thromboxane B(2) in age-related diseases: Focus on atherothrombosis. Ageing Res Rev. 2018;48:51-78.

3.Were the samples collected 24 h after the last aspirin intake?

Response: We thank the reviewer for their careful attention to the timing of sample collection. In response to the query, we confirm that urine samples for the measurement of 11dhTxB2 were indeed collected 24 hours after the last aspirin intake. This timing was deliberately chosen to ensure a standardized assessment of aspirin's antiplatelet effect. By collecting samples at this interval, we could evaluate thromboxane suppression at a consistent time point following the final dose. Given that aspirin irreversibly inhibits COX-1, the 24-hour post-dose collection ensures that the observed 11dhTxB2 levels reflect the cumulative impact of the seven-day treatment period.

To make this detail explicit and address the reviewer’s concern, we have updated the revised manuscript with the following clarification in the Methods section (Page [7], Lines [148-150]):

"Urine samples for 11dhTxB2 measurement were collected 24 hours after the last aspirin intake to standardize the timing of sample collection relative to the final dose."

We believe this addition enhances the transparency of our methodology and aligns with established practices in studies evaluating aspirin’s pharmacodynamic effects. We appreciate the reviewer’s diligence in raising this point, as it has helped us strengthen the manuscript.

4.Was the drug given in the range of 81-100mg according to body weight or was some specific dose given to patients?

Response: Thank you for your query regarding the aspirin dosage in our study. To address your question directly: all patients received a fixed daily dose of 100 mg of low-dose enteric-coated aspirin for seven consecutive days. The dosage was not adjusted based on body weight or any other individual characteristics. Instead, a specific dose of 100 mg per day was administered uniformly to all participants.

To provide further context, the dosage range of 81-100 mg mentioned in the study reflects the commonly accepted therapeutic range for aspirin’s antiplatelet effects. In this case, we selected a fixed dose of 100 mg/day as the standard for all participants, which aligns with typical clinical study designs evaluating aspirin’s inhibition of platelet COX-1 activity in adults. This fixed-dose approach eliminates the need for weight-based adjustments, as the therapeutic effect is achieved consistently within this dosage range across adult populations.

We hope this clarifies the administration protocol used in our study!

5.Provide information about the risk assessment based on levels of 11-dehydro-TXB2.

Response:Thank you for your valuable suggestion. We acknowledge the importance of elaborating on the risk assessment associated with levels of urinary 11dhTxB2 in the context of our study. Below, we have provided additional information based on the data presented in our manuscript and supported by relevant literature, which has been integrated into the revised manuscript under the “Discussion” section.

In our study, diabetic patients exhibited significantly higher baseline urinary 11dhTxB2 levels (median: 2324.58 pg/mg creatinine, IQR: 1447.50–3401.08) compared to healthy controls (median: 1452.51 pg/mg creatinine, IQR: 819.70–2302.12; P = 0.001). Post-aspirin intervention, 11dhTxB2 levels remained elevated in the diabetic group (median: 1205.31 pg/mg creatinine, IQR: 877.52–1446.69) compared to controls (median: 1058.02 pg/mg creatinine, IQR: 641.41–1252.66; P = 0.007). These persistently elevated levels suggest a prothrombotic state in diabetic patients, even after aspirin therapy, which aligns with the observed 2.7-fold higher prevalence of biochemical aspirin resistance (20.8% vs. 7.6%, P = 0.036).

The clinical relevance of 11dhTxB2 levels as a risk marker for cardiovascular events has been well-documented. Eikelboom et al. (2002) demonstrated that elevated urinary 11dhTxB2 levels are strongly associated with an increased risk of myocardial infarction, stroke, or cardiovascular death in high-risk patients, independent of traditional risk factors such as hypertension, hyperlipidemia, or smoking [1]. Specifically, their study reported a dose-response relationship, with higher quartiles of 11dhTxB2 correlating with incrementally greater cardiovascular risk (hazard ratio: 1.8 for the highest vs. lowest quartile, P < 0.001). Although our study did not directly assess clinical outcomes, the elevated baseline 11dhTxB2 levels in diabetic patients (60% higher than controls) suggest a heightened thrombotic risk profile, consistent with this prior evidence.

Furthermore, post-aspirin 11dhTxB2 levels provide insight into aspirin’s efficacy in mitigating this risk. Our threshold for biochemical aspirin resistance (≥1500 pg/mg creatinine) aligns with consensus criteria [2], and the 20.8% resistance rate in diabetic patients indicates a substantial subset with inadequate thromboxane suppression. Persistent 11dhTxB2 levels ≥1500 pg/mg creatinine post-therapy have been linked to a 2- to 4-fold increased risk of adverse cardiovascular events in prior studies [3]. For instance, McCullough et al. (2017) found that patients with stable coronary artery disease and elevated 11dhTxB2 had higher all-cause mortality (adjusted HR: 1.67, 95% CI: 1.03–2.71, P = 0.038) [4]. While our study focused on biochemical resistance rather than clinical endpoints, these findings imply that diabetic patients with post-ASA 11dhTxB2 levels ≥1500 pg/mg creatinine (approximately 20.8% of our cohort) may face elevated cardiovascular risk, warranting closer monitoring or alternative therapeutic strategies.

To enhance the risk assessment discussion in our manuscript, we have revised the “Discussion” section as follows:

"The elevated baseline urinary 11dhTxB2 levels in diabetic patients (2324.58 vs. 1452.51 pg/mg creatinine

---

## [Decision Letter · Decision Letter 1]

28 Jul 2025

Dear Dr. Chen,

Thank you for submitting your manuscript to PLOS ONE. After careful consideration, we feel that it has merit but does not fully meet PLOS ONE’s publication criteria as it currently stands. Therefore, we invite you to submit a revised version of the manuscript that addresses the points raised during the review process.

=====

I was not the original Academic Editor for this manuscript but was assigned after three reviewers weighed in on the revised manuscript. While they are satisfied with the author's responses I strongly believe that the authors need to address several additional points, predominantly with respect to the interpretation of the data. These include:

1. The use of the phrase "factors influencing aspirin resistance" in the title suggests a causal link between the examines variables and the entity referred to as "aspirin resistance." This type of analysis cannot determine causality, it can only investigate an association between the risk variables and the outcome. I would strongly urge the authors to change the title, perhaps to "Factors associated with.." and edit the text to remove any claim of causality.

2. The term "aspirin resistance" is often used but I would ask the authors to reflect on and perhaps clarify to what it actually refers. In the broadest sense, it may mean the failure of aspirin to prevent recurrent cardiovascular events. In the narrowest sense, and the one that many experts in the field prefer, it means the failure of aspirin to inhibit platelet thromboxane generation. There is a large body of evidence to suggest that aspirin is very effective at inhibiting platelet thromboxane generation, even in diabetics, and the authors should review these studies. In that sense, "aspirin resistance" defined narrowly is rather rare. The authors use the measurement of thromboxane metabolites in the urine to assess "aspirin resistance." At the time the assay was developed, it was thought that that all thromboxane in the body was produced by platelets and therefore elevated urinary thromboxane metabolites in aspirin users reflected failure of aspirin to inhibit platelet thromboxane generation. Studies that utilized assays reflecting both platelet-specific and total body thromboxane generation using urine metabolite measurement revealed that non-platelet sources of thromboxane generation account for the overwhelming majority of total body thromboxane in compliant aspirin users, even those with diabetes. The reason is thought to be due to the fact that aspirin has a very short half-life (~20 min) and while it inhibits platelet COX-1 for the life of the platelet, nucleated cells can regenerate COX-1 in a few hours after a once daily aspirin dose, or as the authors correctly point out make TXA2 from a COX-2 pathway. The statement that " urinary 11-dehydrothromboxane B2, the primary TXA2 metabolite, directly reflects platelet-derived TXA2 production (line 88) is simply not true. Rather, urine TXA2 metabolites in aspirin non-users reflects both platelet and non-platelet thromboxane generation and in aspirin users reflects predominantly thromboxane generation from non-platelet sources. The truth is, no one really knows how much thromboxane is made from platelet versus non-platelet sources in individuals not on aspirin with systemic diseases such as diabetes. The authors should revise their language to reflect these important nuances.

3. The authors did not measure and consider oxidative stress as a variable in their analysis. While they touch on this briefly under Limitations, they should expanding this discussion of this limitation. Oxidative stress has been shown in multiple studies by several groups to be one of the strongest risk factors for thromboxane generation. For example, in a study of aspirin users with documented complete suppression of platelet TXA2 generation, our group found oxidative stress was the most important variable contributing to elevated urinary TXA2 metabolites (i.e. non-platelet TXA2 generation) that was directly associated with increased thrombosis and mortality (PMID 27068626, 29097390). Similar results were found in the largest study to date to investigate the association of urinary TXA2 metabolites and outcome (involving over 3000 Framingham participants), where oxidative stress was a stronger risk factor for elevated urinary thromboxane levels than diabetes and inflammation (PMID 35660296)

4. The authors should be aware that the 1500 pg/mg creatinine threshold for determining aspirin responsiveness was derived from the manufacturer of the assay and not based on any outcome analysis but rather on comparison to results from a first-generation assay whose threshold was determined arbitrarily (see PMID 38416712 for an outcome analysis of different thresholds for this assay).

We look forward to receiving your revised manuscript.

Kind regards,

Jeffrey J. Rade, MD

Academic Editor

PLOS ONE

Journal Requirements:

Reviewers' comments:

Reviewer's Responses to Questions

**Comments to the Author**

Reviewer #1: All comments have been addressed

Reviewer #2: All comments have been addressed

Reviewer #3: All comments have been addressed

2. Is the manuscript technically sound, and do the data support the conclusions?

Reviewer #1: Yes

Reviewer #2: Yes

Reviewer #3: Yes

3. Has the statistical analysis been performed appropriately and rigorously?

Reviewer #1: Yes

Reviewer #2: Yes

Reviewer #3: Yes

4. Have the authors made all data underlying the findings in their manuscript fully available?

Reviewer #1: Yes

Reviewer #2: Yes

Reviewer #3: Yes

5. Is the manuscript presented in an intelligible fashion and written in standard English?

Reviewer #1: Yes

Reviewer #2: Yes

Reviewer #3: Yes

Reviewer #1: I am satisfied with the content and revisions provided. The manuscript quality has significantly increased.

Reviewer #2: My concerns and comments have been acceptably addressed. The authors have clarified all methodologic procedures and provided rationale as appropriate.

Reviewer #3: All comments have been addressed by the author. The authors expanded the discussion section which may explain residual thromboxane biosynthesis despite ASA therapy. They also added a discussion on potential collinearity between metabolic-inflammatorymarkers (e.g., BMI and hs-CRP) and subgroup heterogeneity within the DM cohort, which may explain why these factors were not independent predictors in logistic regression but emerged as significant in linear regression. Minor corrections such as lack of consistent font size, lack of significance in the figures have be addressed.

**Do you want your identity to be public for this peer review?** For information about this choice, including consent withdrawal, please see our Privacy Policy

Reviewer #1: No

Reviewer #2: No

Reviewer #3: No

---

## [Author Response · Author response to Decision Letter 2]

26 Aug 2025

Dear Academic Editor :

Thanks for your letter and for the reviewers’ comments concerning our manuscript entitled “Factors Influencing Aspirin Resistance in Diabetic Patients: A Metabolic and Inflammatory Profile Analysis” (ID: PONE-D-25-04526R1). Those comments are all valuable and very helpful for revising and improving our paper, as well as the important guiding significance to our researches. We have studied comments carefully and have made correction which we hope meet with approval. Revised portion are marked in red in the paper. The main corrections in the paper and the responds to the reviewer’s comments are as following:

The academic editor:

1.The use of the phrase "factors influencing aspirin resistance" in the title suggests a causal link between the examines variables and the entity referred to as "aspirin resistance." This type of analysis cannot determine causality, it can only investigate an association between the risk variables and the outcome. I would strongly urge the authors to change the title, perhaps to "Factors associated with.." and edit the text to remove any claim of causality.

Response: We sincerely thank the academic editor for this crucial and insightful comment regarding the interpretation of our study findings. We fully acknowledge and agree that the observational, cross-sectional nature of our case-control design inherently limits our ability to establish “causality” between the examined metabolic-inflammatory variables and biochemical aspirin resistance. The term "influencing" in the original title could indeed imply a direct causal relationship that our data and study design cannot support. We appreciate the academic editor's constructive suggestion to use more precise language that accurately reflects the associative nature of our findings.

Revisions made:

Title Revision:The title has been changed to: "Factors associated with aspirin resistance in diabetic patients: A metabolic and inflammatory profile analysis". This revision explicitly frames the study as investigating associations.

Comprehensive Text Revision:We have conducted a thorough review of the entire manuscript (Abstract, Introduction, Discussion, and Conclusion) to identify and modify any language that might inadvertently imply causation. Key terms implying causality (e.g., "influence," "promote," "drive," "determine," "effect," "leads to," "contributes to" [when implying direct causation]) have been systematically replaced with terms explicitly denoting association or correlation (e.g., "associated with," "related to," "linked to," "correlates with," "found in," "exhibited by"). We have meticulously ensured that the conclusions drawn strictly reflect the observed “associations” within the limitations of the study design.

Emphasis on Study Design Limitation: We have reinforced statements in the Discussion and Limitations sections explicitly stating that our study design identifies associations but cannot establish causation. Specifically, the sentence "Second, the cross-sectional design precludes causal inference..." in the Limitations section now serves as a clearer anchor for this point.

We believe these revisions significantly improve the accuracy and clarity of our manuscript by precisely reflecting the associative nature of our findings, as appropriately constrained by our study design. We are grateful to the academic editor for highlighting this important aspect, which strengthens the scientific rigor of our presentation.

2. The term "aspirin resistance" is often used but I would ask the authors to reflect on and perhaps clarify to what it actually refers. In the broadest sense, it may mean the failure of aspirin to prevent recurrent cardiovascular events. In the narrowest sense, and the one that many experts in the field prefer, it means the failure of aspirin to inhibit platelet thromboxane generation. There is a large body of evidence to suggest that aspirin is very effective at inhibiting platelet thromboxane generation, even in diabetics, and the authors should review these studies. In that sense, "aspirin resistance" defined narrowly is rather rare. The authors use the measurement of thromboxane metabolites in the urine to assess "aspirin resistance." At the time the assay was developed, it was thought that that all thromboxane in the body was produced by platelets and therefore elevated urinary thromboxane metabolites in aspirin users reflected failure of aspirin to inhibit platelet thromboxane generation. Studies that utilized assays reflecting both platelet-specific and total body thromboxane generation using urine metabolite measurement revealed that non-platelet sources of thromboxane generation account for the overwhelming majority of total body thromboxane in compliant aspirin users, even those with diabetes. The reason is thought to be due to the fact that aspirin has a very short half-life (~20 min) and while it inhibits platelet COX-1 for the life of the platelet, nucleated cells can regenerate COX-1 in a few hours after a once daily aspirin dose, or as the authors correctly point out make TXA2 from a COX-2 pathway. The statement that " urinary 11-dehydrothromboxane B2, the primary TXA2 metabolite, directly reflects platelet-derived TXA2 production (line 88) is simply not true. Rather, urine TXA2 metabolites in aspirin non-users reflects both platelet and non-platelet thromboxane generation and in aspirin users reflects predominantly thromboxane generation from non-platelet sources. The truth is, no one really knows how much thromboxane is made from platelet versus non-platelet sources in individuals not on aspirin with systemic diseases such as diabetes. The authors should revise their language to reflect these important nuances.

Response: We sincerely thank the academic editor for this exceptionally insightful and critical comment regarding the precise definition of "aspirin resistance" and the interpretation of urinary 11dhTxB2 levels, particularly in the context of non-platelet sources. We appreciate the opportunity to clarify these crucial nuances, which significantly strengthen the scientific accuracy and interpretation of our findings.

Actions Taken in Response:

Clarified Definition of Aspirin Resistance: We have explicitly defined the specific type of "aspirin resistance" investigated in our study as "biochemical aspirin resistance" throughout the manuscript (Abstract, Methods, Results, Discussion, Conclusion). We emphasize that this definition is based “solely” on persistent thromboxane biosynthesis (urinary 11dhTxB2 ≥ 1500 pg/mg creatinine post-ASA) and “does not imply clinical treatment failure” (i.e., recurrent cardiovascular events). We have added text in the Introduction and Methods sections explicitly stating this distinction and referencing the ongoing debate regarding its clinical relevance. We also acknowledge that true failure of aspirin to inhibit “platelet-specific” COX-1 is likely rare.

Revised Interpretation of Urinary 11dhTxB2: We have completely revised statements regarding the source specificity of urinary 11dhTxB2. Crucially, we have removed the claim that it "directly reflects platelet-derived TxA2 production" (original line 88). We now accurately state that urinary 11dhTxB2 reflects “total body TxA2 biosynthesis”. We explicitly acknowledge that in aspirin-treated individuals, the “majority” of urinary 11dhTxB2 likely originates from “non-platelet sources” (e.g., endothelial cells, monocytes via COX-2 or regenerated COX-1), especially given aspirin's short systemic half-life and the ability of nucleated cells to resynthesize COX. This revision is incorporated into the Introduction (where the biomarker is first described) and extensively discussed in the context of our findings.

Incorporated Discussion of Non-Platelet Sources: We have significantly expanded the Discussion section to address the critical point raised by the academic editor regarding non-platelet sources of TxA2 biosynthesis. We discuss:

The evidence that in aspirin-treated individuals (including diabetics), urinary TxA2 metabolites predominantly reflect non-platelet sources.

The potential mechanisms driving non-platelet TxA2 production in diabetes (e.g., hyperglycemia-induced COX-2 upregulation in endothelial/monocytic cells).

How this understanding contextualizes our finding of persistent 11dhTxB2 elevation post-ASA in diabetics, suggesting it may reflect “systemic (potentially COX-2 mediated) thromboxane overproduction” rather than (or in addition to) true platelet COX-1 resistance.

That the relative contribution of platelet vs. non-platelet sources to baseline (pre-ASA) urinary 11dhTxB2 in individuals with systemic diseases like diabetes remains

Nuanced Language Throughout:We have carefully revised language related to 11dhTxB2 and aspirin resistance across the entire manuscript (Abstract, Introduction, Methods, Results, Discussion) to incorporate these nuances, replacing definitive claims about platelet specificity with accurate descriptions of total body TxA2 biosynthesis and the complexity of its sources, particularly under aspirin therapy.

Re-evaluation of "Resistance" Terminology:While we retain the term "biochemical aspirin resistance" for consistency with established literature using this urinary metabolite threshold, we now explicitly frame it within its limitations: it indicates “inadequate suppression of ‘total body’ TxA2 biosynthesis” by the administered aspirin regimen, the source of which is likely predominantly extra-platelet. We discuss the potential clinical implications (association with cardiovascular risk) despite this lack of source specificity.

We are deeply grateful to the academic editor for highlighting these fundamental aspects of thromboxane biology and aspirin pharmacology. Their critique has led to substantial improvements in the accuracy, depth, and scholarly rigor of our manuscript.

3.The authors did not measure and consider oxidative stress as a variable in their analysis. While they touch on this briefly under Limitations, they should expanding this discussion of this limitation. Oxidative stress has been shown in multiple studies by several groups to be one of the strongest risk factors for thromboxane generation. For example, in a study of aspirin users with documented complete suppression of platelet TXA2 generation, our group found oxidative stress was the most important variable contributing to elevated urinary TXA2 metabolites (i.e. non-platelet TXA2 generation) that was directly associated with increased thrombosis and mortality (PMID 27068626, 29097390). Similar results were found in the largest study to date to investigate the association of urinary TXA2 metabolites and outcome (involving over 3000 Framingham participants), where oxidative stress was a stronger risk factor for elevated urinary thromboxane levels than diabetes and inflammation (PMID 35660296).

Response: We sincerely thank the academic editor for this critically important insight and for highlighting a significant limitation in our study design. We fully acknowledge that oxidative stress is a pivotal driver of thromboxane biosynthesis, particularly from non-platelet sources, and its omission as a measured variable represents a substantial gap in our analysis. We appreciate the academic editor's guidance in directing us to seminal studies, including their own foundational work and the large-scale Framingham cohort analysis, which robustly demonstrate that oxidative stress is a stronger predictor of thromboxane overproduction than traditional risk factors like diabetes or inflammation.

In response to this valuable critique, we have substantially expanded the discussion of this limitation in the manuscript. Specifically, we have:

Added detailed text in the Limitations section explicitly acknowledging oxidative stress as an unmeasured confounder and discussing its established mechanistic role in driving non-platelet thromboxane generation.

Enhanced the Discussion section to contextualize how oxidative stress pathways may underlie our observed associations between metabolic-inflammatory markers (hs-CRP, HOMA-IR) and persistent 11dhTxB2 elevation.

Cited the recommended references (PMID 27068626, 29097390, 35660296) to support these points.

Proposed specific oxidative stress biomarkers and targeted therapeutic strategies in the Future Directions section.

These revisions significantly strengthen the manuscript by providing a more comprehensive perspective on the determinants of thromboxane biosynthesis and aspirin response variability. We are deeply grateful to the academic editor for this expert guidance, which has greatly enhanced the scholarly rigor of our work.

4. The authors should be aware that the 1500 pg/mg creatinine threshold for determining aspirin responsiveness was derived from the manufacturer of the assay and not based on any outcome analysis but rather on comparison to results from a first-generation assay whose threshold was determined arbitrarily (see PMID 38416712 for an outcome analysis of different thresholds for this assay).

Response: We sincerely thank the academic editor for highlighting this important methodological consideration regarding the 1500 pg/mg creatinine threshold for defining biochemical aspirin resistance. We acknowledge that this threshold was primarily derived from the manufacturer's assay validation data and lacks robust outcome-based clinical validation. As appropriately noted, recent evidence (PMID 38416712) demonstrates significant variability in outcome associations across different thresholds for this assay.

To address this limitation, we have:

Revised the Methods section (11dhTxB2 Quantification subsection) to explicitly state the origin of the threshold and its limitation regarding clinical outcome correlation.

Added the suggested reference (PMID 38416712) to support this clarification.

These revisions enhance the transparency of our methodology and appropriately contextualize the interpretation of "biochemical aspirin resistance" prevalence. We believe these changes strengthen the manuscript without altering our core findings.

We tried our best to improve the manuscript and made some changes in the manuscript. These changes will not influence the content and framework of the paper. And here we did not list the changes but marked in red in revised paper.

We appreciate for Editors’ warm work earnestly, and hope that the correction will meet with approval.

Once again, thank you very much for your comments and suggestions.

Yours sincerely

Bo Chen

---

## [Editor Report · Decision Letter 2]

29 Aug 2025

Factors associated with aspirin resistance in diabetic patients: A metabolic and inflammatory profile analysis

PONE-D-25-04526R2

Dear Dr. Chen,

We’re pleased to inform you that your manuscript has been judged scientifically suitable for publication and will be formally accepted for publication once it meets all outstanding technical requirements.

Kind regards,

Jeffrey J. Rade, MD

Academic Editor

PLOS ONE
---

## [Editor Report · Acceptance letter]

PONE-D-25-04526R2

PLOS ONE

Dear Dr. Chen,

I'm pleased to inform you that your manuscript has been deemed suitable for publication in PLOS ONE. Congratulations! Your manuscript is now being handed over to our production team.

Kind regards,

on behalf of

Dr. Jeffrey J. Rade

Academic Editor

PLOS ONE